# Circles of Coastal Sustainability and Emerald Growth Perspectives for Transitional Waters under Human Stress

Ramūnas Povilanskas [1,*], Aistė Jurkienė [2], Inga Dailidienė [3], Raimonds Ernšteins [4], Alice Newton [5] and María Esther Leyva Ollivier [5]

1   Center for Social Geography and Regional Studies, Klaipeda University, 92294 Klaipėda, Lithuania
2   EUCC Baltic Office, 91216 Klaipėda, Lithuania; aiste.jurkiene@gmail.com
3   Marine Research Institute, Klaipeda University, 92294 Klaipėda, Lithuania; inga.dailidiene@ku.lt
4   Department of Environmental Science, Faculty of Geography and Earth Sciences, University of Latvia, LV-1004 Riga, Latvia; raimonds.ernsteins@lu.lv
5   ARNET-CIMA, Universidade do Algarve, 8005-139 Faro, Portugal; anewton.ualg@gmail.com (A.N.); estherollivier87@gmail.com (M.E.L.O.)
*   Correspondence: ramunas.povilanskas@gmail.com; Tel.: +370-615-71-711

**Abstract:** Emerald Growth is an overarching sustainable development framework for transitional waters situated between rivers and open sea. The emphasis on connectivity and ecosystem-based management as the underlying principles differentiates Emerald Growth from conventional approaches to managing transitional waters. The study's primary objective was to conjoin the Emerald Growth concept with the Coastal Circles of Sustainability methodology, an analytical framework to assess indicators of critical processes determining the sustainability of the coastal zone. We hypothesized that applying the CCS is an apt approach to categorizing the Emerald Growth's aspects using Lake Liepāja, a fresh-to-brackish water lagoon on Latvia's Baltic Sea coast, as a case study. Based on the document scoping findings on Lake Liepāja's hydrology, ecology, biodiversity, nature conservation, and management, we addressed the knowledge gaps through the field survey, 4 workshops, and 18 in-depth semi-structured interviews with local stakeholders. The research results show that the challenging socio-economic situation is a crucial obstacle to Emerald Growth in the Lake Liepāja area. Subsistence salary and Housing affordability (Economic Welfare aspect), Population growth and Aging population (Demographic aspect), and Traditional practices (Identity aspect) received the lowest sustainability score (Bad). The results imply that considering the Emerald Growth conditions and drivers for transitional waters worldwide, finding a 'one-fits-all' recipe to ensure their sustainability is impossible. The decision-makers, stakeholders, and external experts agreed that for Lake Liepāja, the priority was to bring back to nature part of the polder system, clean the bottom sediments from Soviet-era pollutants, and enhance the transboundary cooperation with Lithuania. These measures would set the right conditions for future Emerald Growth in the area.

**Keywords:** Baltic Sea; circles of coastal sustainability; coastal lagoons; Emerald Growth; transitional waters

## 1. Introduction

Our research focuses on transitional waters (TW). This physical and ecological domain is crucial for the sustainability and well-being of the coastal regions [1]. According to EU legislation, the term TW refers to coastal water bodies that are partly saline due to proximity to the sea but are influenced by freshwater flows [2]. River estuaries (e.g., the Hudson River estuary or the Thames estuary) and coastal lagoons (e.g., Venice Lagoon) are typical examples of TW. They are globally diverse and highly productive areas, essential for their ecosystem services, such as providing spawning habitats for fish or migration corridors and breeding areas for waterfowl [3]. TW are simultaneously vulnerable and resilient. It makes them ecologically unique [4].

The process of categorizing TW, initiated by the EU Water Framework Directive (WFD, 2000/60/EC), requires consistent typology across the coastal areas [5–7]. Since TW are a continuum lacking criteria that can easily categorize them, they are excluded from the EU Marine Strategy Framework Directive (MSFD, 2008/56/EC) [8]. This has resulted in each coastal Member State of the European Union (EU) adopting different approaches. Some have not designated TW as a coastal water body category [9,10]. Hence, their consistent designation would facilitate cooperation to manage TW sustainably [11].

TW suffer more impacts, resulting in a worse ecological status than lakes and coastal waters [12]. The most stressful European TW concerning human pressures are along the Baltic and North Sea coastlines [13]. Human activities heavily impact TW. Many of them are the sites of major cities and ports [14]. Estuaries, particularly, receive many pollutants from the rivers entering them and land runoff [15]. Pollution negatively affects the ecological robustness of TW habitats [16]. Dredging, aquaculture, and fishing also contribute to the degradation of TW [17–19].

In the EU, the management of TW is complicated due to the transboundary nature of drainage areas [20]. It needs a political commitment to sustainability and coordination of efforts among the EU Member States [21]. The EU prioritizes sustainable growth, promoting Green Growth for terrestrial environments and Blue Growth for marine environments [22,23]. Green Growth fosters economic growth on land while ensuring natural resources are used sustainably [24]. Similarly, Blue Growth deals with the sustainable use of aquatic resources while mitigating marine environment degradation, overuse, and pollution [25].

However, the vagueness of the geographical delimitation of TW presents a challenge when discussing Green and Blue Growth since these concepts are closely interrelated in TW, where terrestrial and marine ecosystems interact. To manage these environments effectively, we proposed the Emerald Growth concept [26]. It combines the principles of Green and Blue Growth to better describe the management of ecosystem services in the continuum between terrestrial and marine areas [27]. This concept relies on sustainable development tenets tailored to TW's unique physical and ecological characteristics [28].

Emerald Growth offers a new perspective on studying and managing TW areas. It addresses sustainable development and management issues within a continuum of upstream catchment areas, downstream TW, and adjacent marine ecosystems [26]. The emphasis on connectivity and ecosystem-based management as the underlying principles differentiates Emerald Growth from conventional approaches to managing TW. It is essential for the EU due to its maritime geography, with 34 of the world's 114 transboundary TW shared between European countries [12].

However, the ecosystem services that form the basis of Emerald Growth must be better defined, since their sustainable use in such complex environments is complicated [29–31]. Regarding TW's connectivity and ecosystem-based management, Emerald Tourism is a priority sector of Emerald Growth. It is a complex activity that combines green and blue tourism features. Analysts of global trends in post-pandemic tourism development emphasize the significant growth potential of environmentally friendly and sustainable tourism in TW areas [32–34]. It aligns with the Emerald Tourism concept.

Our study's primary objective was to combine the Emerald Growth concept with the Coastal Circles of Sustainability (CCS) methodology. It is an analytical framework to assess indicators of critical processes determining the sustainability of the coastal zone [35]. We hypothesized that applying the CCS is a suitable approach to investigate and interpret Emerald Growth's aspects. Therefore, the research aims were as follows:

1. To test the conjoining of the Emerald Growth concept with the CCS methodology using the Lake Liepāja lagoon as a case study.
2. To elicit the main bottlenecks of Emerald Growth in and around Lake Liepāja.
3. To discuss future application perspectives of the Emerald Growth and CCS in TW of the EU and worldwide.

This paper is organized as follows to present the study's results and discuss their implications: it begins with an introduction to the study area and a definition of the survey rationale. A discussion of methods follows. In Section 3, we analyze the parameters defining the CCS and Emerald Growth situation in and around Lake Liepāja. The discussion focuses on the Emerald Growth perspectives for TW bodies under human stress.

We conclude that to achieve Emerald Growth for Lake Liepāja, it is essential to improve the cross-border cooperation with Lithuania in the transboundary Bārta River basin, bring back to nature part of the polder system and take measures of cleaning the bottom sediments from the Soviet era pollutants. We also conclude how the novel insights of conjoining the Emerald Growth concept and the CCS methodology may improve TW management by acknowledging that finding a 'one-fits-all' recipe to ensure TW sustainability is impossible.

## 2. Materials and Methods

### 2.1. Circles of Coastal Sustainability

Emerald Growth implies balancing human and ecological sustainability in TW management and spatial planning [27]. According to the 1987 UN Brundtland Commission Report "Our Common Future", the three pillars of sustainability are (1) Environment and Ecology, (2) Social and Culture, and (3) Economy [36]. We added a fourth pillar, Governance and Policy, to assess the sustainability of management measures. The fragmented governance is one of the main obstacles to sustainable development [37]. This is particularly true for TW, where connectivity and complexity are the issues [38].

We applied the Circles of Coastal Sustainability (CCS) framework developed in 2020 [35] for TW. It includes four interdependent domains (Environment and Ecology, Social and Culture, Economy, and Governance), each with five categories. We used locally adapted indicators to assess each category using a five-grade scale of 1 (Bad) to 5 (Excellent). The results show which domains and categories need priority measures to achieve a sustainable TW system and reinforce environmental resilience. They depend on the specifics of the investigated TW, its catchment area, and the adjacent marine area.

The categories of the four domains are generic sustainability qualities sensitive to a range of scales, from local to regional to national [37]. Choosing indicators reflecting a 'real-world' situation is essential to achieving an ecosystem-based integrated assessment. For Lake Liepāja as a peri-urban managed nature reserve, we emphasized Environment and Ecology (indicators 1. Landscape Alteration; 2. Ecosystem Function; 3. Global Change; 4. Hydrodynamics; 5. Chemical and Physical Flows) in relation to Governance (indicators 1. Organization; 2. Power representation; 3. Law and justice; 4. Legitimacy and accountability; 5. Resource management). Such an assessment framework differs from a TW in a pristine coastal wilderness.

The CCS design aims to achieve a comprehensive evaluation that leads to integrated management, which is vital to Emerald Growth. The graphical representation of the CCS makes communication between stakeholders from different backgrounds and sectors easier [35]. It displays how the deterioration of ecosystem services can impair the sustainability conditions and how power relations, contradictions, and conflicts may influence TW management and facilitate (or not) Emerald Growth. We presented the sustainability assessment for each domain as a collection of spider web diagrams.

### 2.2. Study Area

The study area is in Northern Europe, on Latvia's Baltic Sea coast (Figure 1). It comprises the Lake Liepāja lagoon, the Trade Canal connecting the lagoon with the Baltic Sea, and the watershed of the lagoon, including the Bārta River estuary and the lower stream of the Ālande River. It is one of Northern Europe's largest lagoon lakes [39].

Lake Liepāja is the fifth-largest lake in Latvia, shared between Liepāja City and the South Courland municipalities (Figure 2). The average area of the lake is 37.15 km$^2$, the average depth is 2 m, the maximum depth is 3 m, and the shoreline length is 44.6 km. The drainage area of Lake Liepāja is 2580 km$^2$, with the Bārta River, whose basin is 2016 km$^2$,

being the largest tributary. Lake Liepāja is separated from the sea by a few-kilometer-wide sand bar (Figure 3a). Its water salinity ranges from 5‰ in the northern part (Trade Canal) to 0‰ in the southern part (Bārta River estuary). Lake Liepāja is a eutrophic lagoon with nutrient surplus and extensive areas of emergent vegetation (*Phragmites*, *Typha*, *Scirpus*, *Sparganium*, Figure 3b).

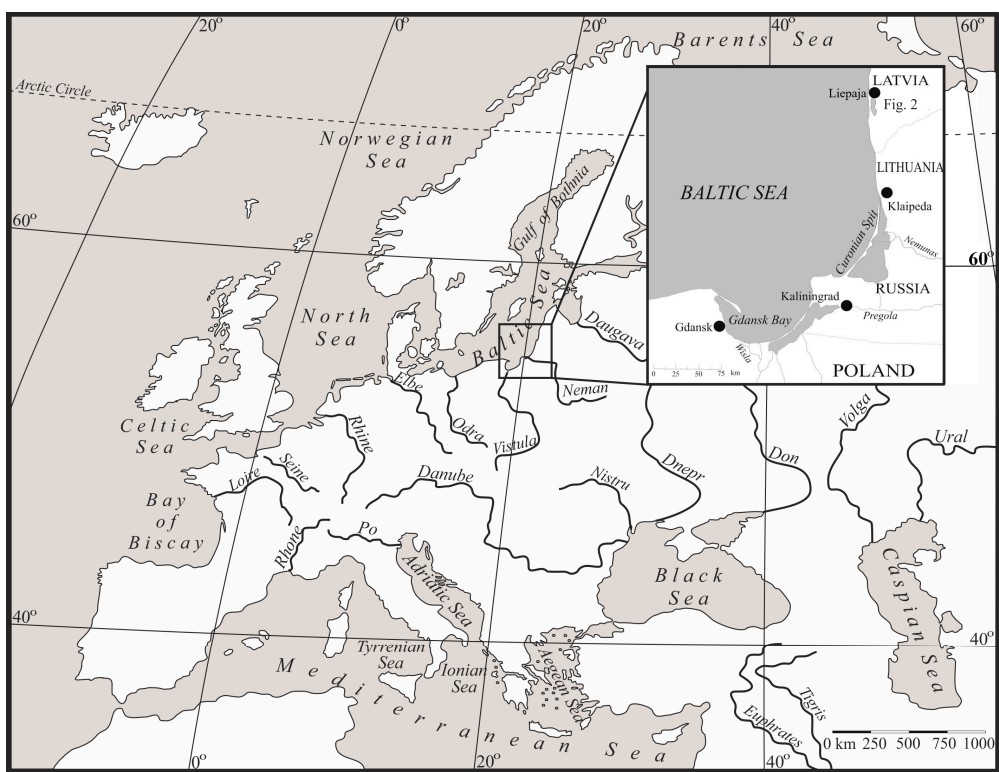

**Figure 1.** Lake Liepāja lagoon is in Northern Europe, on the eastern coast of the Baltic Sea and in the southwestern corner of Latvia. (Image: Ramūnas Povilanskas).

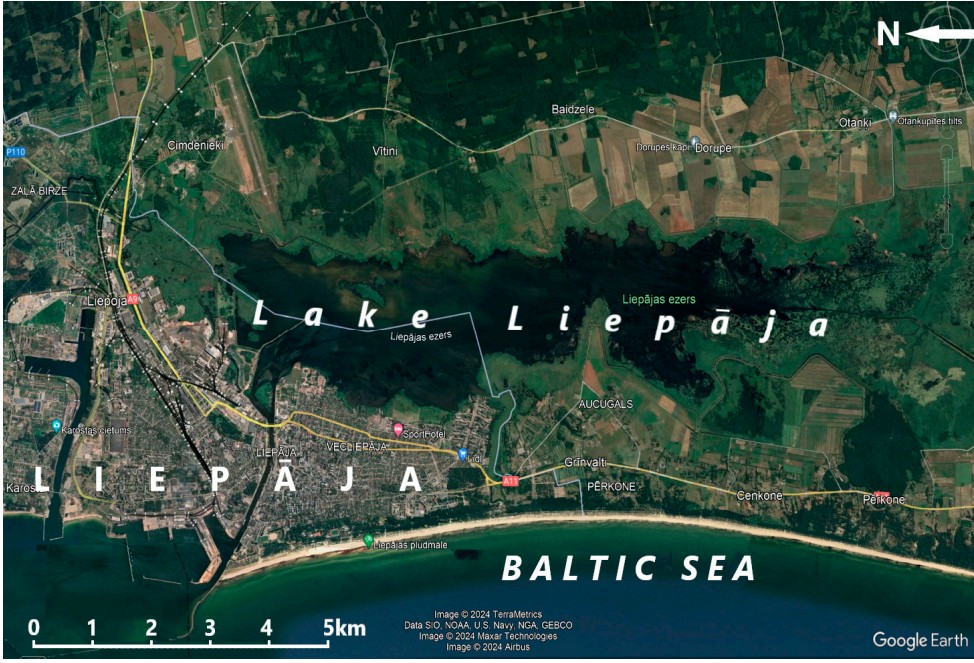

**Figure 2.** Two municipalities of Latvia share the Lake Liepāja lagoon: Liepāja City Municipality and South Courland Municipality. Satellite Image Credit: © Terra Metrics, 2024, Maps Google™.

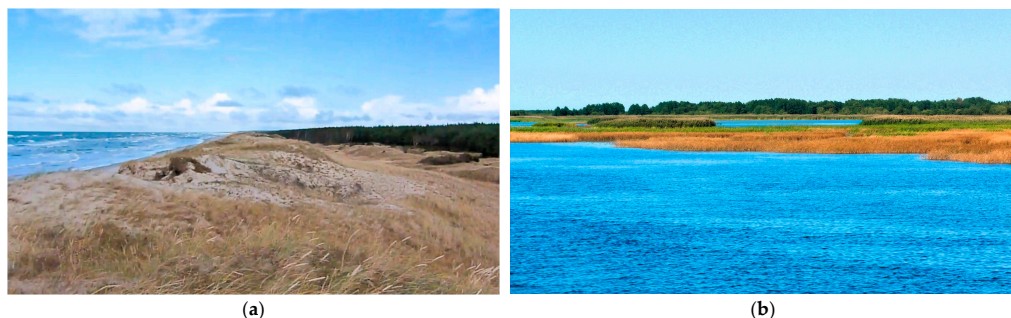

|  (a) | (b) |

**Figure 3.** The sand bar landscape with an artificially created coastal foredune and coniferous Scots' pine forest plantations (**a**) separating Lake Liepāja (**b**) from the Baltic Sea.

The lagoon is in the Bārtava lowland of the Baltic Sea coast [40]. The lowland is on a bedrock depression, descending westwards [41]. Lake Liepāja acquired its current shape during the Littorina Sea, a Baltic Sea transgression 7500–4000 years BP at its peak covering 26.5% more land than today [42]. Since then, coastal erosion, accretion, and bottom sedimentation have been the main geomorphological processes in the area. The lagoon bed is covered with a 0.4–1-m-thick silt layer [43].

Lake Liepāja is in the temperate Atlantic climate coastal region, strongly influenced by the breeze circulation, causing relatively sunny and balmy weather [44]. July is the warmest month in the current period of the climatic standard norm (1991–2020). Its average air temperature is +17.8 °C. February is the coldest month, with an average air temperature of −1.1 °C. Winters are mild, with unstable snow cover and a frost-free period of 140 to 150 days [45]. The ice covers the lake incompletely and sporadically [46]. This allows many waterfowl to winter here.

The lagoon's immediate watershed is in the Baltic Seacoast Hydrological District [47]. According to the hydrological regime indicators, it has the second-highest precipitation indicators in Latvia, relatively high runoff indicators (6th place in Latvia), and the highest evaporation rates in Latvia. The hydrological balance characteristics of the Baltic Seacoast Hydrological District are given in Table 1 [48].

**Table 1.** Indicators of the Baltic Seacoast Hydrological District's hydrological regime.

| Indicator | Annual Amount, mm | Share from Precipitation (%) |
|---|---|---|
| Precipitation | 801 mm | -- |
| Surface runoff | 254 mm | 32% |
| Evaporation | 547 mm | 68% |

Since it is a shallow lagoon with good aeration, its water temperature is closely correlated with the air temperature. It is susceptible to the consequences of climate change, e.g., suffering from more frequent anoxic conditions caused by more frequent hot weather spells. The key indicators characterizing the morphometry and hydrology of Lake Liepāja are given in Table 2 [43].

**Table 2.** Indicators characterizing the morphometry and hydrology of Lake Liepāja.

| Indicator | Average [1] | Characteristic Water Level Values [2] | |
|---|---|---|---|
| | | −0.6 m NN | +0.5 m NN |
| Area (km$^2$) | 37.15 | 24.5 | 46.5 |
| Volume (Mio. m$^3$) | 74.3 | 11.00 | 85.5 |
| Shoreline length (km) | 44.6 | -- | -- |
| Maximum depth (m) | 3 | -- | -- |
| Average depth (m) | 2 | 0.45 | 2.16 |
| Maximum length (km) | 15 | -- | 15 |
| Maximum width | 3.5 | -- | 3.5 |

[1] [49], [2] [50].

The hydrodynamic regime of Lake Liepāja results from an interaction of two opposite flows: the discharge of fresh water from the tributaries and the influx of brackish water from the sea via the Trade Canal during storm surges. Monthly average water levels are higher in the lake than in the sea (up to 10–15 cm in autumn and winter and 0–10 cm in summer). The water level difference between the sea and the lake may reach 30–40 cm during the spring freshets and 25–30 cm during autumn and winter floods. The key indicators characterizing the hydrodynamic regime of Lake Liepāja are given in Table 3 [51].

**Table 3.** Indicators of the hydrodynamic regime of Lake Liepāja.

| Indicator | Value |
|---|---|
| The maximum discharge from Lake Liepāja with a 1% probability | 450 m$^3$/s |
| The maximum discharge from Lake Liepāja with a 5% probability | 343 m$^3$/s |
| The minimum discharge from Lake Liepāja with an 85% probability | 1.92 m$^3$/s |
| The minimum discharge from Lake Liepāja with a 95% probability | 1.54 m$^3$/s |
| The lowest water level in Lake Liepāja with a 1% probability | –0.5 m (NN) |
| The highest water level in Lake Liepāja with a 1% probability | +1.5 m (NN) |
| The total volume of Lake Liepāja for the average water level | 74.3 Mio m$^3$ |

The water exchange between the Bārta River, Lake Liepāja and the Trade Canal is not regulated. The average annual runoff from Lake Liepāja to the Baltic Sea is 0.873 km$^3$ [46], whereas the average annual discharge of the Bārta River is ca. 0.69 km$^3$ [52], and the water volume in the lake ranges from 0.011 km$^3$ to 0.074.3 km$^3$. Hence, the water retention time in the lake lasts from a few weeks to a month. Due to the shallow depth and the exposure of Lake Liepāja to prevailing westerly winds, there is good vertical mixing and insignificant stratification, except for a few enclosed bays with stagnant water.

Meanwhile, the natural hydrodynamics of the immediate watershed of Lake Liepāja is substantially changed. An extensive system of polders with dikes protecting pastures, meadows, and arable land from flooding and canals draining the surplus water to the lake was created in the 1950s and 1960s, separating many adjacent floodplains from the lake.

In 2023, the total population around Lake Liepāja within its watershed was 75,000 (67,000 living in Liepāja City and the rest in South Courland Municipality) [53]. The population density of Liepāja City is 990/km$^2$. South Courland population density is 10/km$^2$. The study area suffers from rapid depopulation. Between 2000 and 2022, the population declined by 25% [53]. The fertility rate in Latvia was 1.6 in 2022 (158th place in the World) [54]. Our research has shown that it is even lower in the Lake Liepāja area. It is in Latvia's periphery where young people tend to leave.

For the last century, the chemical situation of Lake Liepāja, regarding the water quality, nutrient status, oxygen condition and concentration, heavy metals, etc., was complicated. The bottom sediments are of particular concern [55]. In the second half of the 20th century, *Liepājas Metalurgs*, a heavily polluting and water-consuming metallurgy plant, built a dam to separate the contaminated part from the rest of the lake [56]. Although *Liepājas Metalurgs* stopped its activity in recent years, until now, the northwest part is the most problematic environmental hotspot of the lake [57].

In 1977, Lake Liepāja was designated as a managed ornithological reserve to protect nesting and migrating birds, primarily swans, geese, and ducks (Figure 4). In total, 27 bird species nest in the lake, 10 bird species winter here, and 50 protected bird species feed there during migration. A total of 41 protected plant species are also found around the lake [43]. Thirty fish species are found in Lake Liepāja and the Trade Canal, including freshwater and migratory ones, essential for commercial fishing, angling, and nature conservation [58]. Since 2004, Lake Liepāja has been included in the EU Natura 2000 network as a Natura 2000 birds and habitats directives site ("Liepājas ezers" LV0507500) [59] with eight protected habitats.

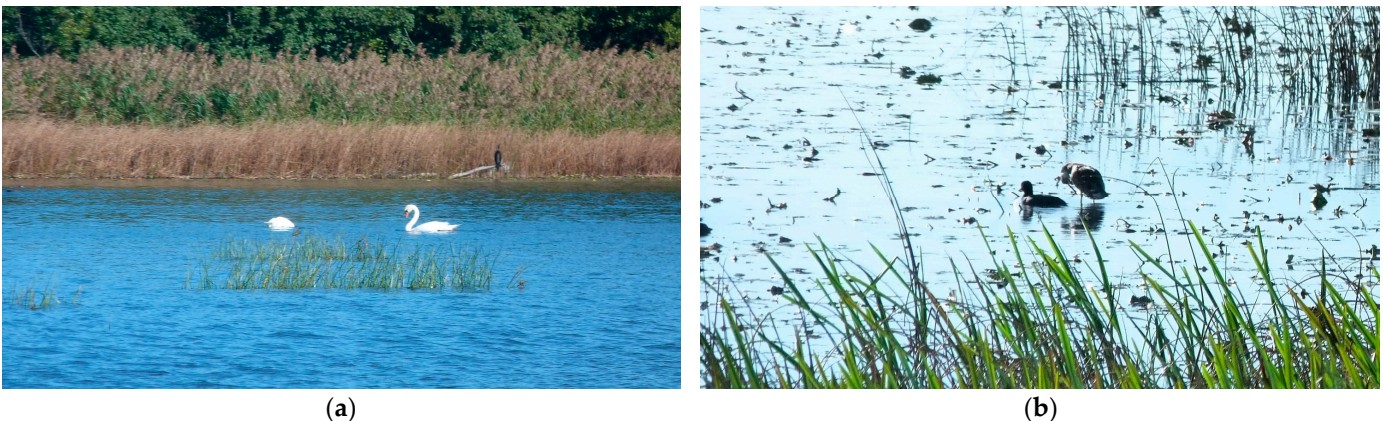

(**a**)                                                                                            (**b**)

**Figure 4.** Protected Natura 2000 aquatic habitats of Lake Liepāja are especially suitable for foraging waterfowl: the mute swan (*Cygnus olor*) (**a**) and duck species (**b**) (Images: Ramūnas Povilanskas).

The lacustrine and riparian habitats, including fish migration corridors, are well-connected except for overgrown parts of the Bārta estuary. Although there were no new dikes, dams, or other barriers erected between 2001 and 2022, the depopulation of the area combined with the environmental problems as a legacy from the Soviet period made us look deeper into the interrelations between the sustainability domain of Environment and Ecology in combination with the Social and Culture domain (indicators 1. Demography; 2. Social benefits; 3. Social well-being; 4. Identity; 5. Social resilience).

### 2.3. Survey Rationale

As part of the implementation process of EU directives–WFD (2000/60/EC), MSFD (2008/56/EC) and MSPD (Maritime Spatial Planning Directive 2014/89/EU), the aquatic environment emerged in the political discourse in the EU Member States. The level of large political systems is crucial if we analyze global trends and propose legislation. Countries, however, are just one frame around human activities, and merely looking at this level is too crude to understand how the TW's aquatic environment and adjacent terrestrial areas are managed. Another possibility is to focus on the primary human activity sectors affecting the TW areas and analyze what people do, how they do it, and what the consequences are. It requires extensive fieldwork.

If we take such a position, the TW entities seen from an environmental or geographical point of view vanish. We are left with several interrelated domains. They rarely communicate or interact directly, but altogether, the consequences of their activities interweave, shaping what we categorize as the quality of the aquatic environment. Our investigations focused on this last approach. We considered four interdependent domains to determine the sustainability level of TW management. For this aim, the CCS framework utilizes a 'dashboard' concept. It allows the summarizing of outputs from diverse sources [35] and from fieldwork.

### 2.4. Data Collection

The paper resulted from our long-lasting dedicated work on this subject. We started the latest study phase in 2021 with a document scoping on Lake Liepāja's hydrology, ecology, biodiversity, nature conservation, tourism development, and governance. We collected data from many sources: research papers, official documents, reports, unpublished materials, other archive documents and literature. Many of these Latvian, English, and Russian documents have been stored at the University of Latvia since the Soviet period. The latest statistical information on the socio-economic and land management situation was obtained from the relevant data sources on the Internet.

Based on the document scoping findings, first we addressed the knowledge gaps through the multi-disciplinary field survey in the Lake Liepāja area. From 2021 to 2023, we

surveyed the study area in detail during our annual international student summer camp fieldwork. The survey was conducted for two weeks every year within the framework of cross-border collaboration between Latvia and Lithuania. It dealt with the regional geography, ecology, hydrology, and socio-economic situation of Lake Liepāja.

We accomplished an in-depth analysis of the document scoping results during a series of four workshops among the authors of this article conducted from 2022 to 2024. The relevant EU directives set the analysis framework: WFD (2000/60/EC), MSFD (2008/56/EC) and MSPD (2014/89/EU). The authors discussed how to enhance Emerald Growth opportunities for Lake Liepāja. We focused on Latvia's maritime spatial planning (MSP) system. The key was the Governance and Policy domain with its five categories and locally adjusted indicators [35]. We used Google Meet© as a platform for the online workshops.

It is accepted that individual in-depth interviews deliver coherent ecosystem service information, as do focus groups [60]. It is possible to follow an exciting line of arguments that only occur during the conversation, and much background information is collected just by 'being there' [61]. Opinions about the environment, ecosystem goods and services, conservation and development conflicts were critical issues during the in-depth individual interviews with the key stakeholders in 2023. We ensured the participants' representativeness regarding gender, age, and interests by sampling the opinions of eighteen interviewees: four municipal decision-makers, two farmers, two commercial fishermen, three anglers, two birdwatching guides, two coastal rural farm owners, and their three visitors, reaching the saturation of the opinions [62,63].

## 3. Results

### 3.1. Circles of Coastal Sustainability of Lake Liepāja: Environment and Ecology

The overall situation with the environment and ecology as the first sustainability domain in Lake Liepāja and its immediate watershed is moderate [Figure 5a]. Lake Liepāja was a typical example of the riven approach to nature conservation and environmental protection in Soviet society. The TW environment and nature were considered resources to be consumed limitlessly. On the other hand, nature, especially on sand bars and barrier spits, was considered a vulnerable treasure to be strictly protected [17]. This split made it meaningful to make a drastic structural interference with natural processes and produce pollution from agriculture, households, and industry, simultaneously maintaining stringent regulations for using and accessing protected areas [61].

In the case of Lake Liepāja, the restrictions were imposed on the ornithological managed reserve and the sand bar. In contrast, the first approach was applied to the northwest corner of Lake Liepāja, where the metallurgy plant was taking fresh water from the lake and discharging wastewater contaminated with heavy metals back [56]. The consequences of this approach are still painful as achieving a Good Environmental Status, according to the EU WFD, is postponed for Lake Liepāja till 2027.

The riparian landscape changes around Lake Liepāja were positive for maintaining biodiversity thanks to implementing the Natura 2000 management plan (Figure 5b). Still, there were negative landscape changes in the wider watershed area as rapid commercial forest felling caused a 25% loss of forest acreage between 2001 and 2022 [64]. Despite habitat management and commercial reed harvesting, the reed bed acreage increased by 30% between 2001 and 2022. This is because the freshwater influx from the Bārta River basin to the lake increased by 12% due to climate change [65].

Trophic and food web interactions in Lake Liepāja are simple, with freshwater northern pike (*Esox lucius*) being the top predator (Figure 5c). However, creating the polder system with dikes around the lake in the 1950s to 1970s eliminated a better part of floodplains, which served as a spawning habitat for northern pike and *Cyprinids*—a process typical for coastal lagoons [66]. Invasions by alien species *Chelicorophium curvispinum* (Sars, 1895) and *Obesogammarus crassus* (Sars, 1894) are observed with unknown distribution and ecological impact on the lagoon [67].

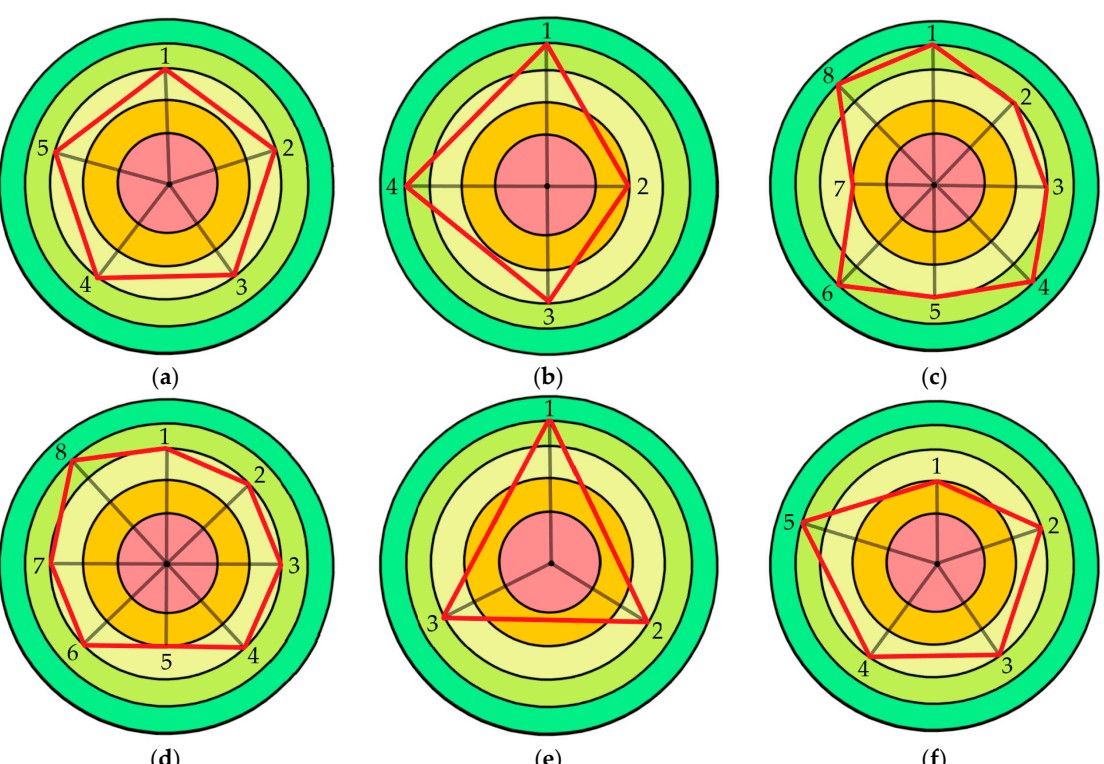

**Figure 5.** Lake Liepāja CCS. Environment and Ecology—Bad (Red color) to Excellent (Green color): Overall (**a**): 1. Landscape alteration; 2. Ecosystem function; 3. Global change; 4. Hydrodynamics; 5. Chemical and physical flows. Landscape Alteration (**b**): 1. Adjacent land change; 2. Catchment land change; 3. Shoreline change; 4. Lagoon change. Ecosystem Function (**c**): 1. Biodiversity; 2. Physical support for biodiversity; 3. Trophic complexity; 4. Keystone species; 5. Alien species invasion; 6. Species conservation; 7. Productivity; 8. Regulation ecosystem services. Global Change (**d**): 1. Sea level change; 2. Salinity change; 3. Water temperature rise; 4. Air temperature rise; 5. Occurrence of heat waves; 6. Thermal regime; 7. Acidification; 8. Increasing coastal erosion. Hydrodynamics (**e**): 1. TW system hydrodynamics; 2. Natural changes in hydrodynamics; 3. Artificial changes in hydrodynamics. Chemical and Physical Flows (**f**): 1. Nutrient cycle; 2. Pollution from agriculture; 3. Other pollutants; 4. Wastewater management; 5. Sediment resuspension.

Lake Liepāja is essential in providing diverse ecosystem services like the regulation function protecting the low-lying Liepāja City from storm surges in the Baltic Sea due to the lake's hydraulic connection with the sea. Extensive and expanding reed beds hinder fish migration [68]. Still, the reed beds serve for carbon storage/sequestration [43]. Despite biodiversity protection, the Lake Liepāja area is susceptible to long-term changes induced by global climate change (Figure 5d). The brackish water zone and related habitats retreat to the north, where the influx of brackish water from the Baltic Sea is more frequent due to increasing storm surge events (Figure 5e).

The average annual air temperature had risen from +6.7 °C from 1961 to 1990 to +7.9 °C from 1991 to 2020 [69]. There were seven heat waves in the summer of 2022 (June–August) on the Latvian Baltic seacoast, with 150 air temperature records broken. The Baltic coastal region witnesses the increasing frequency of heat waves [70]. On the other hand, coastal erosion is not significant yet, even in more frequent storm surges [71].

Algal 'blooms' cause high phosphorus concentrations and sporadic oxygen depletion in the secluded bays of the western part of Lake Liepāja [72]. Regarding $P_{tot}$, the lake's eastern (rural) part shows good ecological quality, whereas the western (urban) part is polluted (Figure 5f). In the catchment area, the most significant diffuse source of pollution is from arable land (64% load of nitrogen and 30% of phosphorus). Discharge from forest land brings 22% of the total nutrient load [51].

### 3.2. Circles of Coastal Sustainability of Lake Liepāja: Economy

The economic sustainability level as the second sustainability domain is relatively good (Figure 6a). In 2023, the average wage before taxes was EUR 1220 per month in Liepāja City and EUR 1160 per month in South Courland [73]. However, working-age people living on an average wage may fall into poverty eight months after losing their jobs and relying solely on unemployment benefits (Figure 6b). Nevertheless, small and medium-sized enterprises (SMEs) in Liepāja City have successfully mitigated the collapse of large-scale industry and provided a wide array of new job opportunities [74].

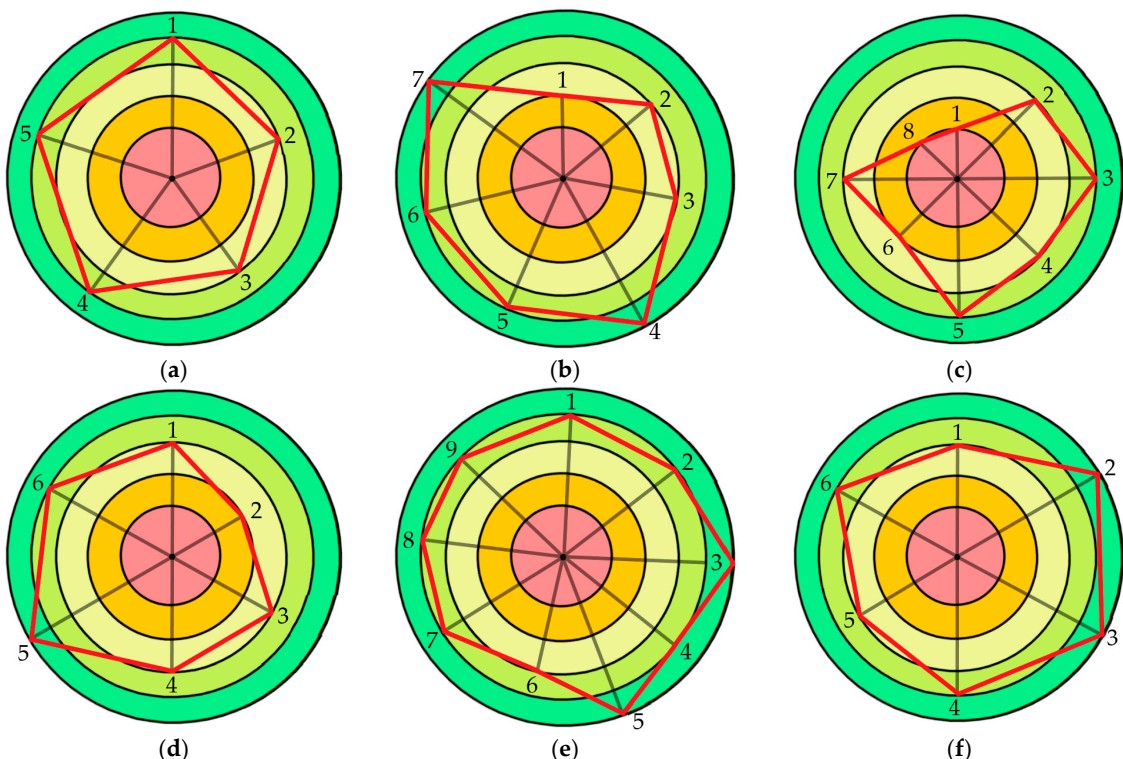

**Figure 6.** Lake Liepāja CCS. Economy—Bad (Red color) to Excellent (Green color): Overall (**a**): 1. Economic security; 2. Economic well-being; 3. Infrastructure; 4. Industry; 5. Reliance on TW. Economic Security (**b**): 1. Poverty risk; 2. Seasonal work; 3. Livelihood; 4. Job opportunities; 5. New economy; 6. Illegal work; 7. Real estate ownership. Economic Welfare (**c**): 1. Subsistence income; 2. Seasonal employment safety nets; 3. Number of working hours; 4. Safe job conditions; 5. Physical labor vs. desk jobs; 6. Social security safety nets; 7. Public service quality; 8. Housing affordability. Infrastructure (**d**): 1. Public transport accessibility; 2. Slow mobility; 3. Ports; 4. Airports; 5. Border crossing points; 6. Energy infrastructure. Industry (**e**): 1. Extractive and non-extractive industry; 2. Water-consuming industry; 3. Fisheries; 4. Farming intensity; 5. Aquaculture industry intensity; 6. Sea transport intensity; 7. Water sports; 8. Leisure industry; 9. RTD sector. Reliance on TW (**f**): 1. Fisheries; 2. Agriculture; 3. Aquaculture; 4. Port services; 5. Leisure services; 6. Hospitality services.

Income inequality in Latvia is moderate (the Gini index was 0.805 in 2021, 64th place in the world) [75]. In 2021, 83.2% of Latvia's inhabitants owned real estate (8th place in the EU) [76]. However, household sale or rent prices are high considering the cost of living [77]. Those who do not own a house or a condo apartment cannot afford to purchase or rent it in Liepāja City (Figure 6c). In 2022, Latvia's average labor hours per worker were 1553 [78], less than the 1570 EU average. Still, the hospitality sector is notorious for overtime working hours in the peak season and minimum wages [79].

The overall infrastructure in the area is moderate (Figure 6d). The Liepāja port served 1652 vessels and transshipped 7 Mio. t of cargo in 2021 [80]. The *Stena* Ro-Pax ferries are working on the Liepāja-Travemünde (Germany) line. Before COVID-19, Liepāja Airport

(LPX) carried 20,000 passengers annually [81]. After the pandemic, airBaltic, the home airline, did not resume regular flights from Liepāja. Palanga International Airport (PLQ) in Lithuania is 60 km from Liepāja. The border with Lithuania is 45 km to the south on the highway A12 Liepāja-Klaipeda. Since both countries belong to the EU Schengen zone, no border or customs control exists except for emergencies.

Liepāja City is rapidly modernizing its industry (Figure 6e). The leading industry branches are those processing various commodities, the textile industry, shipbuilding, and production of automation systems, machines, and metal frames. The Lake Liepāja area is the prime place in Latvia for harvesting wind energy (Figure 7a). It is where wind power is the strongest in the whole country. In 2002, the first wind farm in Latvia was established in Grobiņa, north of Lake Liepāja, with 33 wind turbines (total capacity—9.6 MW) [82].

The total annual landing limit in Lake Liepāja is 2.3 tons for northern pike (*Esox lucius*) and 26 tons for other commercial fish [83,84]. In total, 13 tons of fish were landed in 2019 (including 2 tons of pike), and the trend is declining (Figure 7b) [58]. Farming around Lake Liepāja relies on pumping surplus water out of polders, irrigation when necessary, and greenhouses [82]. The development of water sports in Lake Liepāja is not sustainable. The total number of boats in the lake is over 1000 (Figure 8a). There are 25 boat berths around the lake. Each summer, 250 boats are on the lake, offering commercial services from angling to bird watching. Ca. 2000 anglers use licensed fishing services (Figure 8b).

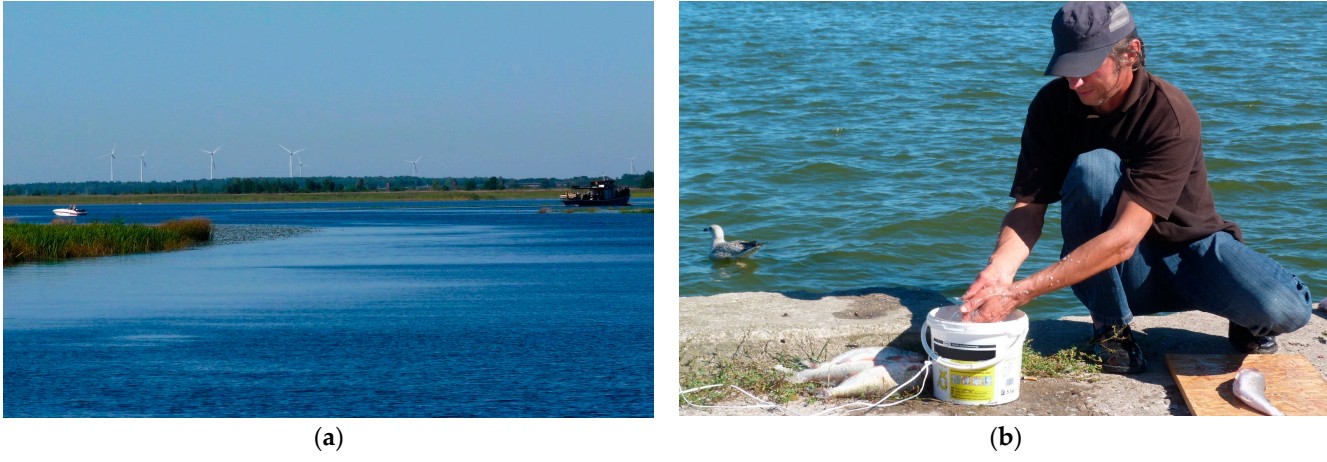

(**a**)　　　　　　　　　　　　　　　　(**b**)

**Figure 7.** Economic activities in the Lake Liepāja area include leisure boating and the wind farms on the northern shore (**a**) and low-scale commercial fishing (**b**) (Images: Ramūnas Povilanskas).

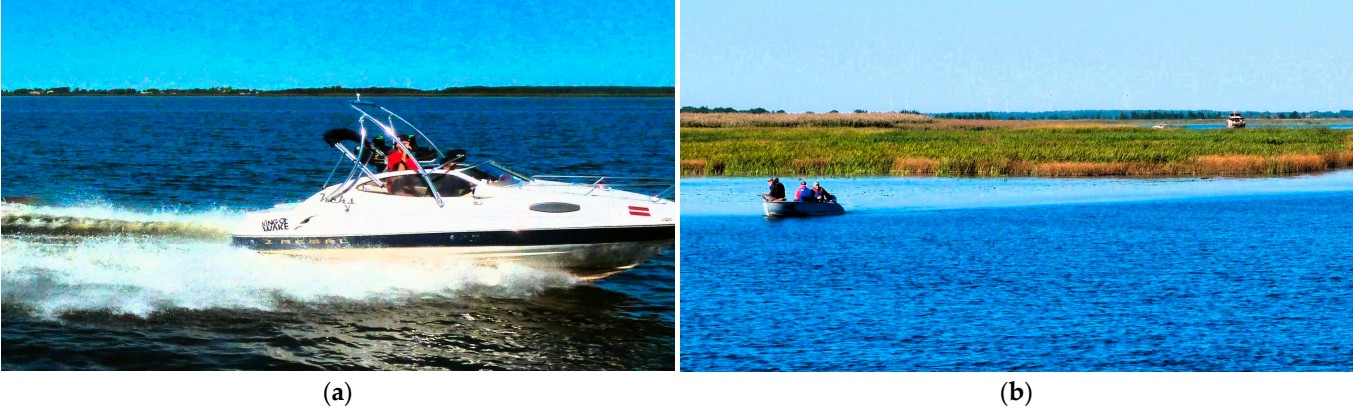

(**a**)　　　　　　　　　　　　　　　　(**b**)

**Figure 8.** Up to 250 leisure boats, including speed boats, use Lake Liepāja at once each summer (**a**), and over 2000 anglers use licensed fishing services annually (**b**) (Images: Ramūnas Povilanskas).

The lake is famous as a hunting place for waterfowl, mainly ducks. There are 1300 hunters in the Lake Liepāja area, which is 7% of all Latvian hunters. In total, 3000 waterfowl are hunted in Lake Liepāja annually, which is 8% of all the waterfowl hunted in Latvia. Also, 3000 birdwatchers visit the area annually. Our research showed a latent conflict between speed boaters and birdwatchers, not between the latter and waterfowl hunters.

In 2021, tourism facilities provided 230,000 overnights for tourists, with peak visitors in July and August [82]. The bathing water quality in Lake Liepāja does not meet the EU Bathing Water Directive (BWD, 2006/7/EC) regulations. There are no officially designated, well-facilitated bathing beaches. An opposite situation is on the Baltic Sea coast of the sand bar. There, the water quality usually meets the BWD regulations. Two municipal seaside beaches in Liepāja City enjoy the Blue Flag award. It is an important quality indicator [85].

The proportion of jobs dependent on the TW ecosystem goods and services is robust (Figure 6f). In 2019, 60 licenses were issued for commercial fishing in Lake Liepāja, 16—to legal entities employing up to 5 fishermen, and the remaining to individuals. Hence, the total number of fishermen in Lake Liepāja is 100, but declining. In total, 2200 people work in agriculture, and 2500 employees work at the port of Liepāja. There are 20 hotels, 60 restaurants and cafes with 2000 employees, and 23 seaside and lakeside rural tourism farms with over 100 mainly seasonal employees around the lake.

### 3.3. Circles of Coastal Sustainability of Lake Liepāja: Social and Culture

The social and cultural sustainability domain is pivotal for the Lake Liepāja area since it is the heart of the Lower Courland ethnographic region. Before the Soviet occupation, the rural areas around Lake Liepāja were the heartland of the *Kurzemnieki* culture of fishing farmers with integrated farming and fishing practices (e.g., collecting beach wrack to enrich their barren soils), traditional clothing and crafts (e.g., cutting reed and thatching roofs, collecting amber on seaside beaches, smoking fish, etc.). However, we now evaluate the area's social and cultural sustainability as moderate (Figure 9a).

As mentioned, the demographic situation in the Lake Liepāja area is very challenging, especially in rural areas (Figure 9b). Also, Liepāja City suffers from population ageing, emigration, and the resulting decline in inhabitants [53]. On the other hand, this trend leads to a more sustainable spatial urban and peri-urban development without any urban sprawling or overcrowding [82]. The ethnic composition is also becoming more balanced compared with the Soviet era [53].

The social amenities of the Lake Liepāja peri-urban area for visitors and inhabitants are plenty (Figure 9c): fresh air, low noise levels, and wide-open water space. The quality of fish landed from Lake Liepāja and pesticide residues in fruits and vegetables grown in the area meet EU food safety regulations. Still, the housing segregation is high (Figure 9d). Next to luxurious seaside condos, there are many low-quality condos or derelict old-block housing districts. Such controversies throughout the country predetermine that Latvia's Happiness index value was 6213 in 2023 (41st place in the World), way below neighboring Lithuania (6763, 20th place) [86].

Individuals and communities of the Lake Liepāja area connect moderately in their identity with the environment (Figure 9e). The Soviet regime made persistent efforts to impose totalitarian collectivism. Therefore, the society's atomization was the prevailing trend as a counter-reaction after Latvia restored Independence in 1990. There was little interest in social connections beyond the immediate neighborhood or hobby groups. Nowadays, the sense of community is getting stronger with the 'gentrification' of society, though the investments in upgrading community centers have been low until now [82]. The fact that Liepāja will become Europe's Capital of Culture in 2027, coinciding with the city's 400th anniversary, also enhances social cohesion.

We evaluated the sense of environmental responsibility and justice of the inhabitants in the Lake Liepāja area as moderate (Figure 9f). Likewise, the population's social resilience to environmental hazards and disasters is moderate. After the upheaval in the late 1980s, the sense of environmental justice subsided due to society's atomization and socioeconomic

challenges. Environmental education, i.e., focusing on environmental awareness, climate change, and resulting environmental hazards, is unpopular. In the academic year 2020/2021, out of 4065 school children attending one of the five off-school interest education programs in Liepāja City and South Courland municipalities, just 9 children chose the environmental education program.

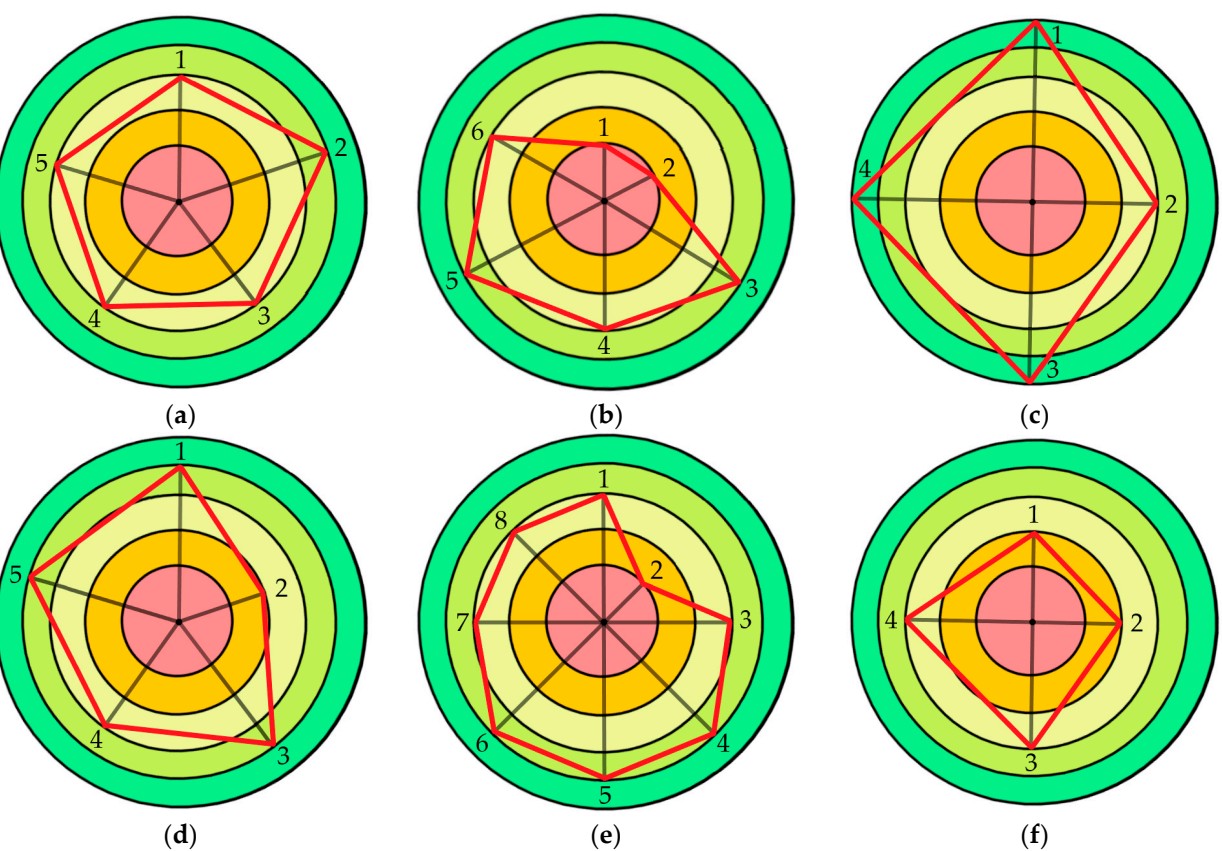

**Figure 9.** Lake Liepāja CCS. Social and Culture—Bad (Red color) to Excellent (Green color): Overall (**a**): 1. Demography; 2. Social amenities; 3. Social well-being; 4. Identity; 5. Social resilience. Demography (**b**): 1. Population growth; 2. Aging population; 3. Population density; 4. Migration and immigration; 5. Visitors and residents; 6. Percentage of minority groups. Social Amenities (**c**): 1. Food provision; 2. Bathing water; 3. Access to natural space and biodiversity; 4. Aesthetic and hedonistic benefits. Social Well-being (**d**): 1. Food quality; 2. Housing segregation; 3. Perceived safety; 4. Daily nature exposure; 5. Recreational activities. Identity (**e**): 1. Sense of community; 2. Traditional practices; 3. Folklore; 4. Cultural participation; 5. Cultural heritage preservation; 6. Social cohesion outside the traditional practices; 7. Sense of environmental responsibility; 8. Sense of environmental justice. Social Resilience (**f**): 1. Education; 2. Social awareness of environmental risks; 3. Social responses to hazards; 4. Availability of disaster insurance.

The communities around Lake Liepāja show an overall negligible level of knowledge and awareness about the preparedness for environmental hazards, as the collapse of the Liepāja municipal wastewater treatment facilities in July 2023 showed. The 'blame game' that followed revealed that the main culprit was an inadequate understanding of the longshore sediment drift under the influence of the Liepāja port jetties. Still, there was a robust social response to the COVID-19 pandemic two years ago. Ad hoc social networks were created, facilitating collective societal actions to assist with quarantine and philanthropy.

### 3.4. Circles of Coastal Sustainability of Lake Liepāja: Governance

We evaluated the overall governance sustainability in the Lake Liepāja area only as moderate (Figure 10a). Community-based management is almost non-existent in environmental protection, except as ad hoc initiatives of the EU-subsidized Local Action Groups. The system's organizational entropy is high. The administrative organization, i.e., administrative competence and cooperative involvement, is challenging for South Courland Municipality. It is a new territorial entity organized on 1 July 2021, out of eight small former municipalities, resulting in the "fragmentation of competence".

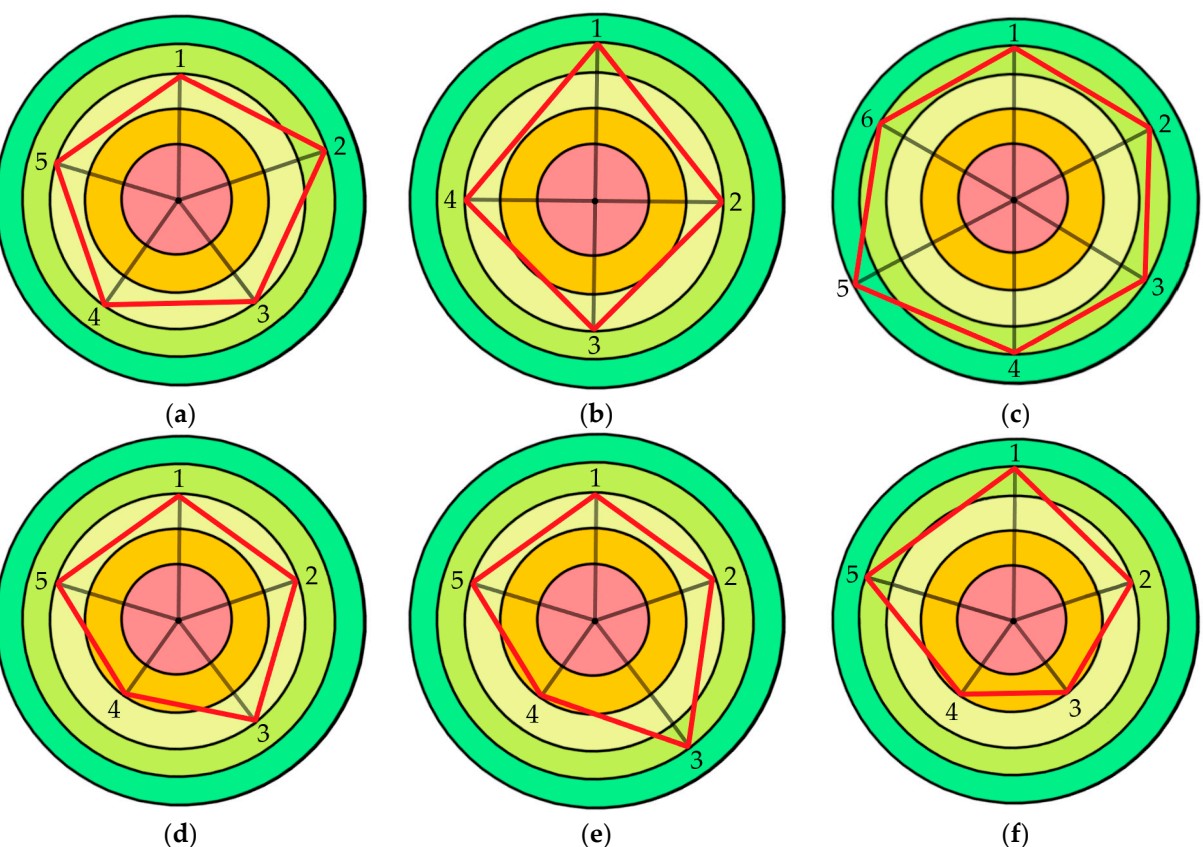

**Figure 10.** Lake Liepāja CCS. Governance—Bad (Red color) to Excellent (Green color): Overall (**a**): 1. Organization; 2. Power representation; 3. Law and justice; 4. Legitimacy and accountability; 5. Resource management. Organization (**b**): 1. Number of organizations around the ecosystem; 2. Organizational maturity; 3. Entropy of the system; 4. Bureaucracy. Power Representation (**c**): 1. Transdisciplinary collaboration; 2. Technical advisory management; 3. Inclusive policy; 4. Participatory approach; 5. Traditional participation; 6. Number of qualified specialists. Law and Justice (**d**): 1. Regulatory environmental policies; 2. Adaptive regulations; 3. Licensing control efficiency; 4. Percentage of illegal actions in the system; 5. Number of lawsuits, fines, or penalties. Legitimacy and Accountability (**e**): 1. Transparent processes and communication; 2. Accountability instruments; 3. Law enforcement; 4. Transparency of accountability penalties; 5. Evidence of government corruption. Resource Management (**f**): 1. Regional partnerships; 2. Community-based management organization; 3. Management tools; 4. Crisis management; 5. Communication and science.

The decision-making process on environmental issues is top-down in both municipalities sharing the Lake Liepāja area (Figure 10b): over 30 institutions deal with environmental protection and nature conservation. Even the NGO responsible for lake management (*Liepājas ezeri*) was established top-down by municipalities sharing Lake Liepāja and other local lakes [87]. The lack of coherence within the municipal environment organization weak-

ens control mechanisms, as the collapse of the wastewater treatment facilities in July 2023 showed. The authorities stopped the spill within 24 h. Still, it caused the contamination of the sea beaches at the peak of the summer seaside tourism season.

The network of institutions with a stake in environmental management is transitioning from an immature to a maturing organization. It has operation guidelines and objectives, but its structure needs improving. The representation of individuals with diverse identities in institutions regarding gender, cultural, or ethnic background is moderate, with ethnic Latvian males prevailing (Figure 10c). Still, both municipalities seek expert opinions to make informed decisions and reinforce system management.

In Latvia, regulations exist for nature conservation, air and water pollution control, climate change mitigations, sustainable urban planning, and land use on the national, regional, and local levels (Figure 10d). However, legally binding regulations take time to modify or change in response to new environmental challenges and situations. For example, the Lake Liepāja Nature Conservation Plan fixed nature conservation regulations and measures for 15 years [43].

The accountability for harvesting ecosystem goods (Figure 10e) and the resource management sustainability in and around Lake Liepāja is moderate (Figure 10f). In 2022, Latvia's corruption perception index value was 54 (39th among the World's countries, much worse than neighboring Estonia, 14th place) [88]. Our research has shown that commercial fishermen and anglers undeclared about 12 tons of landed fish in 2022. In total, 70% of visitors at the rural tourism farms pay in cash and are not subject to taxation. However, the ecological values of Lake Liepāja are well documented and communicated. There are 200+ short videos uploaded on YouTube. Approximately 100+ research papers on the lake's environment published between 2008 and 2023 are accessible via the Google Scholar platform.

## 4. Discussion

The essential difference in nature management approach between the EU management of Natura 2000 protected areas and the Soviet nature conservation comes from different protected area management paradigms–integration in the EU and segregation in the Soviet Union. The EU Birds Directive (BD, 79/409/EEC) and the Habitat Directive (HD, 92/43/EEC) emphasize proactive habitat conservation and management measures. These measures are listed in comprehensive nature management plans. The Latvian nature conservation system is transitioning from the Soviet to the EU nature management paradigm.

A similar case study from a peri-urban Natura 2000 site in another post-Communist country, Poland, emphasizes the need to integrate different policy sectors at regional and local levels [89]. The aim is to create a network of areas delivering a wide range of ecosystem services. Its resilience relies on several aspects: connectivity, multifunctionality, applicability, integration, diversity, multiscale, governance, and continuity [90]. Considering Lake Liepāja as a typical TW area, the focus should also be on environmental flows, water quality, invasive species, integrated water resources management, strategic conservation planning, and emerging ecosystem monitoring technologies [91].

At the turn of the 21st century, a shift in TW floodplain management towards the non-farm sector emerged [11]. The TW fringes were once considered worthless wetlands that were to be reclaimed and converted into agricultural land. Nowadays, they have become appreciated as valuable ecosystems providing unique services [92]. Similarly, restoring highly altered riparian areas was prioritized in preserving aquatic biodiversity [93,94]. Eliminating marginal polders and restoring natural hydrological and ecological processes increases the acreage of valuable dwelling and spawning habitats for fish and waterfowl [11].

This scheme looks nice in theory but is challenging to implement in practice. If economic difficulties in the country arise, nature conservation expenditures are the first to be sacrificed [95]. It is a safer guarantee while restoring the natural watershed of Lake Liepāja to consider the economic benefits of floodplains for their yield in natural products. However, the local rural economy around Lake Liepāja is susceptible to seasonality. This

dilemma is also witnessed in other TW areas [96]. The clue to this challenge lies in shifting the EU Common Agricultural Policy towards the Green Deal, raising support for organic farming, and maintaining biological diversity [97].

However, it is problematic, as the latest Europe-wide farmers' protests against the Green Deal showed [98]. The farmers around Lake Liepāja receive EU subsidies and want to continue extensive farming. Still, some economically viable agroecology examples can be applied in TW areas. Intercropping, agroforestry, and mixed grazing facilitate positive interaction between species and life forms [99]. It creates synergies when producing local products from specific terroirs. The Palavas Lagoon Complex in South France is an excellent example of the terroir-based use of TW with sustainable oyster aquaculture and environmental cattle breeding in the adjacent saltmarshes [100,101].

The issue of rural socio-economic sustainability is closely related to another long-term threat to Emerald Growth, i.e., dramatic depopulation of the area. This threat is hard to mitigate or eliminate. Lake Liepāja is on the country's periphery, where accessibility, facilities, economic conditions, population ageing, natural amenities and the degree of urbanization are at play [102]. The declining population of Liepāja City as a regional urban hub turns the depopulation problem of the Lake Liepāja area into a 'vicious circle'.

Bearing this in mind, the identified stark contrast between the widely spread private household ownership for 'haves' and soaring prices to buy or rent a house or a condo for 'have-nots' in Liepāja City and the adjacent seacoast looks odd. This paradox can be explained by a global feature that TW urban waterfronts in lagoons (e.g., Venice), sand bars (e.g., Miami Beach), and estuaries (e.g., Shenzhen) are among the most appealing areas for real estate developers [103,104].

Liepāja citizens blame the municipality for the paucity of affordable housing in a shrinking city [77]. It brings us to governance sustainability and efficiency. As mentioned above, the organizational entropy of the system is high. It is transitioning from an immature to a maturing one, according to the organizational maturity assessment (OMM). It is a methodology to gauge the maturity level of an organization (or a network of organizations). OMM reflects its capability in terms of management [105], including knowledge, process, and performance management [106].

To enhance organizational maturity, the involvement of local inhabitants in the decision-making process is necessary. Knowledge of natural resources, ecosystem dynamics, and associated management practices exists among local people interacting with ecosystems daily and over extended periods [107]. For instance, 76% of Baltic seaside beach visitors in Liepāja are local inhabitants [108]. They are the most positive in evaluating the condition of their city's beaches among the surveyed visitors in Latvia's four most popular seaside resorts.

Successful models of community engagement in environmental decision-making that can be implemented locally exist. They focus on helping grassroots initiatives like youth participation in international cooperation projects on environmental justice. For instance, the Radi Vidi Pats (Create the Environment Yourself) association motivates young people to become environmental activists. Also, local branches of national environmental NGOs, like the Latvian Fund for Nature, the Environmental Protection Club, and the Latvian Ornithological Society, contribute to conserving Lake Liepāja's biodiversity.

Liepāja City integrates the management of its beaches and water resources into the overall management framework for its TW water bodies and Natura 2000 sites. Liepāja's socioeconomic well-being relies on effectively managing water resources, making the water sector an integral part of the city's overall operation and management [87]. Integrated lake and water resource and coastal area management are vital components of Liepaja City's development documents and spatial plan, which should be consistent and coherent with the MSP of Latvia and the Venta River Basin transboundary management plan [109].

Hence, we must discuss the frameworks to integrate MSP with ecosystem-based TW management. The WFD established the framework for EU action in water policy. It commits EU Member States to achieve a good environmental status of water bodies, including TW

and coastal waters, up to 1 nautical mile (NM) from the coast. The MSFD established a framework for EU action in marine environmental policy to achieve a good environmental status of marine waters. Meanwhile, the MSPD established a framework for MSP in the EU.

To promote the sustainable use of maritime space, MSP must consider land–sea interactions. The MSPD distinguishes between marine and coastal waters. It does not apply to coastal waters under town and country planning. However, Latvia treats the Daugava River estuary and the riverine discharge plume into the Gulf of Riga as a TW. It implies that both WFD and MSPD regulate Latvia's Baltic offshore. We propose to resolve this overlap by designing a special plan for the continuum of TW and coastal waters until 12 NM within a larger national MSP framework. It would ensure connectivity and ecosystem-based management. Germany's North Sea MSP provides a good example of this.

Exploring the broader coherence of our findings to similar TW in other regions, we pinpoint the Zwin estuary, where we conducted research in 2019 [12]. The Zwin is a TW area spanning the border between Belgium and the Netherlands. It is an excellent example of cross-border collaboration in TW management. The International Zwin Commission was established in 1872 to maintain the area jointly. It was designated as a Ramsar wetland of international importance in 1986, and recently both countries agreed to revitalize the TW. Rewilding the cross-border Willem-Leopold polder has been chosen as the best scenario to introduce ecosystem-based management and restore connectivity with the sea.

Despite its quality, our investigation suffered from some biases. For instance, in 18 semi-structured interviews with local stakeholders, we ensured socio-economic representativity and reached the saturation of the opinions [62,63]. Still, we missed the representatives of ethnic minorities living in the area. Ethnic minorities in Liepāja City comprise 41% of the population. In South Courland Municipality, they comprise 11% of the population [53]. Also, in 2022, 2300 Ukrainian war refugees settled in the area whom we also missed. Acknowledging this gap is the main lesson for future research, especially in ethnically more diverse TW areas.

Hence, future research should take the minority voices among the TW stakeholders into account. Also, we need to refine the concept of Emerald Growth by clarifying its links with the EU Green and Blue Growth strategies. Pursuing a novel management approach to address TW's unique challenges includes looking for win-win solutions to promote agroecology and sustainable rural tourism in TW [110]. More attention should be paid to fine-tuning and applying weights to the indicators. Also, there is room for expanding the indicator list to address the United Nations Sustainable Development Goals for the continuum of TW more adequately.

## 5. Conclusions

The primary conclusion is that the results of our investigation show it is feasible and reasonable to combine the Emerald Growth concept with the CCS methodology. The ranking of the domains and categories reveals that the best situation regarding the conditions for Emerald Growth in the Lake Liepāja area is with the economy (three categories—Economic security, Industry, and Reliance on TW—evaluated as good). Also, we evaluated Social amenities (Social and Culture domain) and Power representation (Governance domain) as good. All other domains and their categories received moderate scores.

However, this moderate picture hides the main stressors creating pressure on Lake Liepāja as a TW under human stress. The challenging socio-economic situation is a crucial obstacle to Emerald Growth in the area. The combination of deficiencies in economic welfare (Economy domain) and demographic challenges (Social and Culture domain) pose a particular threat. We assigned the lowest score (Bad) to the following aspects: Subsistence salary compared to minimum wage (Economic Welfare aspect), Housing affordability (Economic Welfare aspect), Population growth (Demography aspect), Aging population (Demography aspect), and Traditional practices (Identity aspect).

We further conclude that to achieve Emerald Growth for Lake Liepāja, it is necessary to improve the cross-border cooperation with Lithuania in the transboundary Bārta River

basin, bring back to nature part of the polder system and take measures of cleaning the bottom sediments from the Soviet era pollutants. These measures can build a synergy to upgrade the worst-evaluated categories of the Environment and Ecology domain (Catchment land change, Productivity, and Nutrient cycle) and the Governance domain (Management tools and Crisis management).

These conclusions show that the challenges for advancing Emerald Growth in the Lake Liepāja area are primarily related to overcoming the Soviet legacy, which is still hideously felt in many ways. A few successful initiatives exist in similar contexts to overcome the Soviet legacy and improve Emerald Growth conditions. For instance, our research in the Nemunas River Delta at the Curonian Lagoon, one of Europe's largest TW, has shown that the rewilding of peripheral polders saves the water pumping and dyke maintenance costs, strengthens the area's appeal for ecotourists and opens new agroecology development perspectives [110].

This article presented the results of the first pilot study to gauge the applicability of the CCS methodology for TW areas and its coherence with the Emerald Growth concept. It will be further tested in a wider TW variety to verify its local adaptability while providing a global overview of TW sustainability. However, it is already evident from the results of this case study that finding a 'one-fits-all' recipe to ensure TW sustainability is impossible. Each TW area must address local economic, ecological, social, and governance priorities and challenges. Adding additional categories and domains to the CCS list may be necessary to analyze other TW with another set of physical and socio-ecological features.

**Author Contributions:** Conceptualization, R.P. and A.J.; methodology, A.N. and M.E.L.O.; software, M.E.L.O.; validation, R.E. and A.J.; formal analysis, R.P. and A.J.; investigation, R.P., A.J. and R.E.; resources, R.E.; data curation, I.D.; writing—original draft preparation, A.J.; writing—review and editing, A.N.; visualization, A.J.; supervision, R.P.; project administration, M.E.L.O. All authors have read and agreed to the published version of the manuscript.

**Funding:** This research received no external funding.

**Institutional Review Board Statement:** The study did not require ethical approval.

**Data Availability Statement:** Data are contained within the article.

**Acknowledgments:** We appreciate the insightful comments and suggestions from the anonymous reviewers that helped substantially improve the quality and presentation of the manuscript.

**Conflicts of Interest:** The authors declare no conflict of interest.

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
