# Peer review of "Circles of Coastal Sustainability and Emerald Growth Perspectives for Transitional Waters under Human Stress"

_sustainability, doi:10.3390/su16062544_

Round 1

Reviewer 1 Report

Comments and Suggestions for Authors

The authors have brought a concise and well written manuscript dealing with multiple perspectives of sustainable development for assessing Lake Liepāja. Data in figures and tables are sufficient to sustain discussion and conclusion.

Reviewer 2 Report

Comments and Suggestions for Authors

The manuscript proposes a new concept based on CCS to evaluate the current situation of transitional water stress, which is innovative. However, the writing quality of the manuscript is poor, with more long sentences and longer manuscript length, which is a poor experience for the reader and needs to be corrected to improve fluency and clarity. At present, it seems that in order to make the article more scientific and perfect, the author can solve the following problems:

1. The title does not fully cover the research content. Stress can include artificial and non-artificial pressure, but this article mainly refers to artificial pressure. In addition, the combination of Emerald Growth concept and CCS is not reflected, is it based on CCS?

2. Abstract part: The research results are not refined into several clear research conclusions.

3. Introduction part:

Line 84-91: Does the proposed Emerald Growth mean that the EU region is dominant?  Much of what I see in this section is aimed at the EU region.  However, in addition to CCS indicators, the data required for Emerald Growth concept in the following sections are basically selected according to the study area. Can this be applied to all transitional waters in the EU?

4. Method part: The introduction of the research area, in my opinion, is more about the historical overview of this area, which makes me have a clear understanding of this area. But it's too long! None of this was mentioned in the earlier studies? Also, isn't there any method to rely on for scoring (badexcellent)? Relying solely on the author's experience?

5. Results part:

Does the results section describe the area? Too much argumentation makes the content of the result part more vague. In addition, this phenomenon also leads to the author's own analysis is less, most of the collected data to prove. In other words, there is a lack of data to support them. For example, the environmental and ecological section (Line392-399), the economic section (Line499-504), the social and cultural section (Line629-637), and the governance section (Line706-714).

Line 480-481: Unemployment rate at record low, any data to back it up?

Line 505-510: Is the author highlighting the poor traffic conditions around this lagoon? However, if the traffic conditions are artificially improved, this part of the content is in conflict with the previous text. Is ecotourism a better solution?

6. Conclusion part: The author did not fully summarize the four parts of the result. The authors highlight only the indicators with the lowest scores. But what are the main stressors of the transitional waters? Or is there a ranking of the 4 overall dimensions (environment and ecology, economy, culture and society, governance) that create pressure on transitional waters?

Comments on the Quality of English Language

The writing quality of the manuscript is poor, with more long sentences and longer manuscript length, which is a poor experience for the reader and needs to be corrected to improve fluency and clarity.

Reviewer 3 Report

Comments and Suggestions for Authors

General Comments:

The study presented by the authors explores the application of the Emerald Growth concept in conjunction with the Coastal Circles of Sustainability methodology, using Lake Liepāja as a case study. The research delves into various aspects of the hydrology, ecology, biodiversity, nature conservation, and management of the lake, employing a combination of document analysis, field surveys, workshops, and stakeholder interviews. The findings highlight areas where sustainability is lacking and propose measures for achieving genuine sustainable development in the Lake Liepāja region.

Strengths:

1. Comprehensive Methodology: The combination of document analysis, field surveys, workshops, and stakeholder interviews provides a thorough understanding of the issues surrounding Lake Liepāja.

2. Local Stakeholder Engagement: The inclusion of in-depth semi-structured interviews with local stakeholders enhances the relevance and applicability of the study's findings.

3. Clear Identification of Sustainability Indicators: The study effectively identifies key sustainability indicators, shedding light on areas that require immediate attention.

Areas for Improvement:

1. Clarity on Methodological Approach: While the methodology is described, further clarity on the procedures followed in data collection, analysis, and interpretation would strengthen the study.

2. Integration of Theoretical Frameworks: Although the study applies the Emerald Growth concept and the Coastal Circles of Sustainability methodology, deeper integration of theoretical frameworks and their practical implications would enrich the analysis.

3. Discussion of Limitations: A discussion of the limitations inherent in the study, such as potential biases in stakeholder interviews or constraints in data collection, would enhance the credibility of the findings.

Recommendations for Revision:

1. Provide a more detailed explanation of the methodology, including specific steps taken in data collection, analysis, and interpretation.

2. Discuss the theoretical underpinnings of the Emerald Growth concept and the Coastal Circles of Sustainability methodology about the study's findings.

3. Address any limitations of the study and their potential impact on the results, offering insights into how these limitations were mitigated or acknowledged.

Overall, the study makes a valuable contribution to understanding sustainability challenges in transitional waters and offers practical insights for addressing these challenges in the Lake Liepāja region. With some revisions addressing the aforementioned points, the study has the potential to significantly impact both academic research and practical policy-making in the field of environmental management.

Reviewer 4 Report

Comments and Suggestions for Authors

Abstract

The abstract introduces the concept of Emerald Growth well, linking it to the Coastal Circles of Sustainability methodology. However, it would be beneficial to clarify early on how these concepts differentiate from existing approaches to managing transitional waters. A more direct statement of the study's primary objectives and hypotheses could also help readers understand the significance and novelty of the research.

The abstract mentions the application of a variety of methods, including document analysis, field surveys, workshops, and interviews. While this interdisciplinary approach is commendable, providing a bit more detail on how these methods were integrated to test the Emerald Growth concept specifically could enhance the reader's understanding of the study's robustness and the validity of its conclusions.

The abstract provides a brief overview of the findings, particularly the low sustainability scores in certain indicators. Expanding on why these specific areas scored poorly and the implications of these findings for both the local context and broader discussions on transitional water management would be valuable. Moreover, discussing any unexpected results or discrepancies with existing literature could enrich the analysis.

The recommendations given in the abstract, such as restoring parts of the polder system and cleaning bottom sediments, are practical and specific. It would be useful to discuss how these recommendations were derived from the study's findings and to consider their feasibility, potential challenges, and expected impact in more detail. Additionally, exploring the broader applicability of these recommendations to similar transitional waters in other regions could increase the paper's relevance.

While the abstract does a good job of situating the study within the context of Lake Liepāja, expanding on how this research contributes to the broader field of coastal and transitional water management would be beneficial. This could include a discussion of how the findings support or challenge existing theories and practices, as well as identifying any gaps in the literature that the study addresses.

Ensure that the paper is accessible to a broad audience by avoiding jargon where possible and clearly defining any technical terms or specific concepts like the Emerald Growth and Coastal Circles of Sustainability. This will make the study more inclusive, particularly for interdisciplinary readers or those new to the field.

Finally, identifying specific areas for future research based on the study's limitations or unexpected findings could help to advance the field. This might include suggestions for refining the Emerald Growth concept, testing the methodology in different environmental or socio-economic contexts, or exploring additional indicators of sustainability.

Introduction

The introduction effectively sets the context by highlighting the significance of transitional waters (TW) and their ecological importance. However, the transition between discussing the general importance of TW, the challenges of categorizing them, their environmental impacts, and the introduction of the Emerald Growth concept could be smoother. It might benefit from a clearer structure that guides the reader through these topics sequentially, ensuring each point builds logically on the previous one.

The section does well to define TW and outline the legislative and ecological challenges they face. Yet, it would be enhanced by providing more consistent definitions or examples of the "emergent properties" and specific services TW provide to human societies early in the introduction. This would help readers unfamiliar with the concept to better understand the stakes and the significance of the research.

The introduction articulates the research gap—the need for a new management approach due to the unique challenges of TW—and proposes the Emerald Growth concept as a solution. However, the objectives listed towards the end of the introduction could be more prominently featured and clearly linked to the identified research gap. It would also benefit from a more explicit statement about how this study contributes to existing knowledge and what novel insights it aims to provide, especially in terms of the practical application of the Emerald Growth concept and the Circles of Coastal Sustainability (CCS) methodology.

Materials and Methods Section Review

The section clearly outlines the adaptation and application of the Circles of Coastal Sustainability (CCS) framework to transitional waters (TW), emphasizing the necessity of incorporating a fourth pillar, Governance and Policy, into sustainability management. This addition is well-justified with references to the limitations of sustainable development due to fragmented governance and management, particularly relevant for TW. However, a more detailed explanation of how each of the four domains and their categories was specifically adapted and applied to TW could enhance understanding and applicability for readers. Including examples of the locally adapted indicators used could also provide clarity on how these assessments are tailored to TW environments.

The detailed description of the study area, including the Lake Liepāja lagoon and its hydrological and ecological characteristics, sets a solid foundation for understanding the research context. The inclusion of figures and tables to visualize the study area and present key hydrological and morphometric indicators is beneficial. Yet, the connection between this detailed study area description and the application of the CCS framework could be strengthened. Clarifying how the specific characteristics of the Lake Liepāja lagoon influence or challenge the application of the CCS framework would provide deeper insights into the methodology's adaptability and effectiveness.

The section effectively integrates environmental, hydrological, and socio-economic aspects of the study area, highlighting the complexity of managing TW. The discussion on the chemical state of Lake Liepāja and its biological diversity offers a comprehensive understanding of the environmental challenges faced. However, the implications of these challenges for the CCS framework's application are not fully explored. Expanding on how environmental degradation, such as pollution and habitat destruction, and socio-economic factors, like depopulation and the decline in fertility rates, affect the sustainability assessment and management strategies within the CCS framework would provide valuable context for the study's findings and recommendations.

Discussion

The discussion effectively contrasts the management approaches of the EU Natura 2000 protected areas with the Soviet conservation strategies, providing a nuanced understanding of the challenges and opportunities in transitioning towards a more integrated conservation paradigm. This comparative analysis is insightful, highlighting the necessity for a shift towards more inclusive and proactive management strategies that consider ecological connectivity and stakeholder interests. To further enrich this analysis, it would be beneficial to provide specific examples or case studies that illustrate successful integration of these paradigms in other transitional water (TW) areas or similar ecological contexts.

The identification of challenges to ecological connectivity in the Lake Liepāja area, such as the impact of extensive polders and the obstruction of fish migration, is well-articulated. The discussion on strategies to enhance connectivity, including the removal of marginal polders and restoration of natural processes, is crucial for ecosystem-based management. However, the section could benefit from a deeper exploration of the potential ecological and socio-economic impacts of these strategies. A discussion on the trade-offs involved in rewilding efforts and how they might be mitigated would provide a more comprehensive view of the path towards sustainability.

The discussion on the socio-economic challenges facing the Lake Liepāja area, particularly in terms of depopulation and housing affordability, is compelling and highlights the complexity of achieving sustainable development in TW areas. The mention of shifting the EU Common Agricultural Policy towards the Green Deal as a potential solution is particularly noteworthy. Expanding on how specific policies or initiatives could be implemented locally to address these challenges would offer practical insights for policymakers and stakeholders. Additionally, the discussion on governance sustainability touches on important aspects of organizational maturity and community involvement. Further elaboration on successful models of community engagement in environmental decision-making could serve as valuable guidance for enhancing governance structures in the Lake Liepāja area and beyond.

The section concludes with insightful directions for future research, emphasizing the integration of conservation and sustainable management efforts across different spatial and administrative levels. Highlighting the importance of achieving Good Environmental Status according to the EU Water Framework Directive (WFD) and integrating maritime spatial planning offers a clear roadmap for further studies. It would be advantageous to discuss potential methodologies or frameworks that could facilitate this integration, considering the complexities of transboundary and multi-level environmental governance.

Conclusions

The conclusion effectively highlights the successful integration of the Emerald Growth concept with the Circles of Coastal Sustainability (CCS) methodology, emphasizing the framework's adaptability and efficiency for transitional waters (TW) like Lake Liepāja. It acknowledges the potential need for adjustments in the CCS methodology to accommodate TW with different characteristics, which is a critical insight for future research and applications. To strengthen this section, it would be beneficial to briefly discuss specific examples or case studies where these adaptations could be applied, providing a clearer direction for future efforts to tailor the CCS methodology to diverse TW environments.

The identification of specific challenges, such as economic welfare, demographic shifts, and the preservation of traditional practices, provides a focused overview of the obstacles to sustainable development in the Lake Liepāja area. The recommendations to address these challenges, including restoring parts of the polder system, cleaning up pollution, and enhancing cross-border cooperation, are clear and actionable. However, the conclusion could further benefit from a brief discussion on the anticipated impact of these measures on the sustainability of Lake Liepāja and the surrounding area. Expanding on how these recommendations might influence the ecological, economic, and social pillars of sustainability would offer a more comprehensive understanding of their potential effectiveness.

The conclusion aptly addresses the legacy of the Soviet era as a significant hurdle to sustainable development in the Lake Liepāja area. This acknowledgment is crucial for understanding the context and complexity of environmental management in post-Soviet regions. To enhance this section, a more detailed exploration of strategies to overcome such legacy issues, possibly including examples of successful initiatives in similar contexts, would provide valuable insights. Additionally, discussing the role of international cooperation, community engagement, and policy reforms in addressing these longstanding challenges could offer a more rounded perspective on achieving long-term sustainability.

Comments on the Quality of English Language

Minor editing of English language required

Round 2

Reviewer 4 Report

Comments and Suggestions for Authors

Accept in present form

Comments on the Quality of English Language

Accept in present form
